# *Juniper communis* L. Essential Oils from Western Romanian Carpathians: Bio-Structure and Effective Antibacterial Activity

Eugenia Dumitrescu [1], Florin Muselin [1], Carmen S. Dumitrescu [2], Sergiu A. Orasan-Alic [1], Răzvan F. Moruzi [1], Alexandru O. Doma [1], Erieg A. Mohamed [1,3] and Romeo T. Cristina [1,*]

1  Departments of Pharmacology and Pharmacy, Toxicology, Faculty of Veterinary Medicine, Banat's University of Agricultural Sciences and Veterinary Medicine, 300645 Timisoara, Romania; eugeniadumitrescu@usab-tm.ro (E.D.); florinmuselin@usab-tm.ro (F.M.); s.orasan@yahoo.com (S.A.O.-A.); moruzirazvanflorin@yahoo.com (R.F.M.); alexandru.doma@usab-tm.ro (A.O.D.); erieg_1980@yahoo.com (E.A.M.)
2  Department Systems in Agriculture, Faculty of Management, Banat's University of Agricultural Sciences and Veterinary Medicine, 300645 Timisoara, Romania; carmen02dumitrescu@gmail.com
3  Department of Applied Sciences, Division of Biotechnology, University of Technology, Baghdad 19006, Iraq
*  Correspondence: romeocristina@usab-tm.ro

**Abstract:** The antibacterial activity of four bacterial standard strains that are naturally encountered in humans and animals was investigated by using bioactive compounds from commercial essential oils of *Juniperus communis* that were collected from the Western Romanian Carpathians. The *Juniper communis* essential oils, volatile compounds, were recognized through the GC–MS methodology by comparing identified spectra with those held in the NIST 02, Wiley 275 library. The ratio of each component was calculated based on the peak areas of the GC, without utilizing correction factors. The CLSI standardized micro-dilution was used to determine antimicrobial activity, employing $10^{-3}$ dilutions of fresh culture, with inoculums equivalent to a standard of 0.5 McFarland being prepared for testing. Four bacterial strains, *Staphylococcus aureus* (ATCC 25923), *Escherichia coli* (ATCC 25922), *Pseudomonas aeruginosa* (ATCC 27853), and *Streptococcus pyogenes* (ATCC 19615), were investigated, using 96-well micro-dilution plates. Over each micro-dilution well, the essential oils were poured, introducing gradually 2, 4, 8, and 10 µL/well, respectively. The results were expressed as ±SEM and analyzed by one-way ANOVA with Bonferroni's multiple comparison test, considering the differences statistically provided when $p < 0.05$ or lower. The juniper essential oil originating from the Western Romanian Carpathians is rich in four main volatile compounds: β-pinene (34.02%), 1α-pinene (30.43%), p-cymol (20.25%), and β-myrcene (10.20%). The juniper communis essential oil reduced bacterial density for all four strains tested, but compared to Gram-negative bacteria, in our case; a considerably higher antibacterial effectiveness was detected for Gram-positives, with peak reduction of *Staphylococcus aureus*, recommending the Romanian essential oil as a beneficial antibacterial resource.

**Keywords:** bioactivity; bio-structure; *Juniper communis*; Carpathian areal

## 1. Introduction

*Juniper communis* is a coniferous belonging to the genus *Juniperus* in the family *Cupressaceae*. In Romania, the species grows throughout the Carpathian chain at 700–1400 m altitude and is often findable in bushes and patches, in meadows, and in pastures, and can grow on the poorest soils [1].

The leaves are linear and sharp at the tip and grouped three vertically at the same level; two kinds of flowers arise on the branches of the second year. The female flowers are spherical, consisting of three whorled carpel scales, each carrying an egg, and the male flowers are ovoid, yellow, with numerous stamens. The fruits—more precisely, the pseudo-fruits—are globose and short-stemmed. The vegetable product used consists of ripe fruits (*Fructus juniperi* or *Baccae juniperi*) [2].

This coniferous plant, since ancient times, has been reported to be multifariously effective in traditional medicine as an diuretic, anti-inflammatory [3–5], antifungal [4,6], analgesic [7], hepatoprotective [8], antidiabetic and antilipidemic [9,10], antimicrobial [11], and antioxidant [4,12]; it also works against nontuberculous mycobacteria [13–15], is neuroprotective in Parkinson's disease [16], and is even cytotoxic against human neuroblastoma cells [4].

As bioactive compounds, fruits contain apigenin, rutin, luteolin, quercetin-3-o-arabinosil-glugoside, quercetin-3-o-rhamnoside querrein, scutellarein, nebetin, ofutavone, and bilobetine [5,17–19], and oil of juniper fruit is composed mostly of monoterpene hydrocarbons, such as β-pinene (5%), α-pinene (51.4%), sabinene (5.8%), myrcene (8.3%), and limonene (5.1%) [20].

The seeds and fruits of the plant contain d-α-pinene, camphene, pectin, glycolic acid, malic acid, formic acid, acetic acid, cyclohexitol, terpenes, proteins, fermentable sugars, wax, gum, ascorbic acid, dihydrogen, β-pinene, hydro-junene, cadinene, juninene, junipere, and camphor [5,20,21].

Limited information can be found about the analysis of the essential oil extracted from juniper berries, and only little information is found on the chemical structure of commercially available oils [20,22,23]. For example, Filipowicz et al. [22] and Gordien et al. [24] ascertained that *J. communis* commercially available essential oils were active against *Escherichia coli* and *Staphylococcus aureus,* with the exception of *Pseudomonas aeruginosa,* which appears to be resistant to *J. communis.*

Hence, the interest in the potential of using juniper essential oils, as several have been shown to have antimicrobial activity against a broad spectrum of bacteria, juniper EO from the diverse geographical origins, including Algeria, Croatia, Estonia, Greece, Italy, Kosovo, Lithuania, Macedonia, Poland, Portugal, Serbia, Spain, Slovakia, and Turkey, illustrated a qualitative and quantitative difference, and further research in this domain may contribute as a valuable database for the future [5,11,17,18,23,25–33].

In this aim, our study proposes biochemical research on the antibacterial activity on four standard strains most commonly found in humans and animals of the main bioactive compounds from the essential oil of *Juniperus communis* provided from the Western Romanian Carpathians (WRC). This investigation comes to fulfill the lack of data about specific main components and structures found in junipers and their specific antibacterial activity, since there are no available data from our country in the mainstream yet.

## 2. Materials and Methods

*Juniperus communis* essential-oil-conditioning Herbalsana (batch no. 1985/20) was purchased from Herbavit SRL, Oradea, Bihor County, RO. According to the producer certificate, the oil was provided from Western Romanian Carpathians (46.75534 N, 22.79702 E) in year 2020, and it was obtained through woody-parts steam-distillation.

As organoleptic proprieties, the essential oil was a clear yellowish liquid of medium viscosity with a rustic, woody, slightly spicy aroma.

### 2.1. GC–MS Identification of Volatile Compounds in J. communis Essential Oil

*J. communis* essential oil analysis was performed by using an Agilent Technology 7820A gas chromatograph (Agilent Scientific, Santa Clara, CA, USA) coupled with MSD 5975 mass spectrometer and equipped with a DB WAX capillary column (30 m × 250 pm × 0.25 pm) and a single quadrupole detector. The gas used was helium, with a mass flow rate of $1 \text{ mL} \times \text{min}^{-1}$. The following oven program was used to separate the compounds: 40 °C for 1 min, $5 \text{ °C min}^{-1}$ to 210 °C for 5 min.

Injector and ion source temperatures were 250 and 150 °C, respectively. The injection volume was 1 μL of each pure mixture or solvent-free oil, with a 1:20 partition ratio; the NIST 2011 (National Institute of Standards and Technology) spectrum library was used to identify the volatile compounds.

A TIC (Total Ion Chromatogram) was created by summing up intensities of all mass spectral peaks belonging to the same scan. The TIC was then compared to the GC chro-

matogram. The identification was made by comparing the obtained mass spectra peaks with those stored in the libraries of NIST 2011. The percentage of individual components was calculated based on the peak areas of the GC without using correction factors (see Supplementary Figure S1).

### 2.2. Determination of Antimicrobial Activity of J. communis Essential Oil

Broth Micro-Dilution Method

The micro-dilution method in broth has been standardized by CLSI (Clinical and Laboratory Standards Institute) for testing bacteria that grow aerobic yeast and filamentous fungi. The micro-dilution method in EUCAST (The European Committee on Antimicrobial Susceptibility Testing) broth is in principle similar to that of CLSI, with changes that usually refer to some of the test parameters, such as inoculums preparation, size, and MIC (Minimum Inhibitory Concentration) reading.

The method is a simple method of antimicrobial-susceptibility testing. The procedure involves testing double dilutions of the analyzed antimicrobial agent in a liquid growth medium distributed in 96-well micro-titer plates. Each well was inoculated with microbial inoculums prepared in the same medium after dilution of the standardized microbial suspension adjusted to the McFarland scale of 0.5.

Thereafter, the 96-well micro-titer plate was incubated under suitable conditions, depending on the microorganism tested, with MIC being the lowest concentration of antimicrobial agent that completely inhibits the growth of the body in the micro-dilution wells.

A $10^{-3}$ dilution of fresh culture with inoculums equivalent to a standard of 0.5 McFarland was prepared and used for testing with the following bacterial strains:

- *Staphylococcus aureus* (ATCC 25923),
- *Escherichia coli* (ATCC 25922),
- *Pseudomonas aeruginosa* (ATCC 27853),
- *Streptococcus pyogenes* (ATCC 19615).

The ATCC strains used were revived by overnight growth in Brain Heart Infusion (BHI) broth (CM1135, Oxoid, UK) at 37 °C and subsequently passed on BHI agar for 24 h at 37 °C. The strains were then diluted to an OD (Optical Density) of 0.5 McFarland standard ($1.5 \times 10^8$ CFU $\times$ mL$^{-1}$), using BHI broth. The resulting suspensions were tested by using a 96-well flat-bottomed micro-dilution plate with a usable volume of 200 μL.

Over each suspension well, the sample was directly poured, introducing 2, 4, 8, and 10 μL into each well. The plates were covered and left overnight at 37 °C; then OD was measured at 590 nm wavelengths, using an ELISA reader (Biorad PR 1100). All tests were performed in duplicate for all samples, with the strain suspensions in BHI being used as a positive control.

### 2.3. Statistical Analysis

The obtained results were expressed as arithmetic mean and SEM (standard Error of the Mean) and were analyzed by one-way ANOVA (Analysis of variance) with the Bonferroni's multiple comparison test considering the differences are statistically provided when $p < 0.05$ or lower, using the GraphPad Prism 6.0 software (GraphPad Software, San Diego, CA, USA).

### 3. Results

Following the gas chromatographic analysis coupled with mass spectrometry, the following compounds were identified in *J. communis* essential oil, as presented in Table 1, and the comparative optical density and statistical values for juniper oil tested for inhibition of *Str. pyogenes, S. aureus, P. aeruginosa*, and *E. coli* are presented in Table 2; and in Supplementary Figure S1, the chromatogram and parameters are presented.

**Table 1.** Main volatile constituents identified in *Juniper communis* L. essential oils.

| Main Compound Found | RT (min) | Concentration (%) | m/z | Area | Height |
|---|---|---|---|---|---|
| 1 α-pinene | 6.397 | 30.43 | TIC | 34,329,052 | 3,325,404 |
| camphene | 7.521 | 0.127 | TIC | 142,832 | 22,109 |
| β-pinene | 8.682 | 34.02 | TIC | 38,379,520 | 3,728,137 |
| norborane | 9.361 | 0.114 | TIC | 128,459 | 25,567 |
| menthene | 9.553 | 0.061 | TIC | 69,123 | 12,698 |
| 3-carene | 9.849 | 2.249 | TIC | 2,537,461 | 354,481 |
| β-myrcene | 10.191 | 10.204 | TIC | 11,511,868 | 1,644,793 |
| α-fenchene | 10.415 | 0.697 | TIC | 785,822 | 129,580 |
| p-cymol | 13.267 | 20.25 | TIC | 22,844,789 | 2,906,183 |
| p-menth-1-en-8-ol | 25.033 | 1.846 | TIC | 2,082,830 | 137,499 |

Note: RT = retention time; TIC = Total Ion Chromatogram.

**Table 2.** Comparative optical density for juniper oil tested for inhibition of *Str. pyogenes*, *S. aureus*, *P. aeruginosa*, and *E. coli*.

| | Conc./replica | I | II | III | $\bar{x}$ | SEM |
|---|---|---|---|---|---|---|
| *Streptococcus pyogenes* | 2 μL | 0.556 | 0.555 | 0.558 | 0.556 | 0.0008 |
| | 4 μL | 0.433 | 0.435 | 0.436 | 0.435 | 0.0008 |
| | 8 μL | 0.354 | 0.355 | 0.356 | 0.355 | 0.0005 |
| | 10 μL | 0.347 | 0.349 | 0.346 | 0.347 | 0.0008 |
| | BHI + Stem. | 0.786 | 0.787 | 0.786 | 0.786 | 0.0003 |
| *Staphylococcus aureus* | 2 μL | 0.109 | 0.110 | 0.112 | 0.110 | 0.0008 |
| | 4 μL | 0.126 | 0.120 | 0.124 | 0.123 | 0.0017 |
| | 8 μL | 0.125 | 0.128 | 0.123 | 0.125 | 0.0014 |
| | 10 μL | 0.126 | 0.122 | 0.130 | 0.126 | 0.0023 |
| | BHI + Stem. | 0.496 | 0.496 | 0.494 | 0.495 | 0.0006 |
| *Pseudomonas aeruginosa* | 2 μL | 0.109 | 0.099 | 0.105 | 0.104 | 0.0029 |
| | 4 μL | 0.131 | 0.129 | 0.137 | 0.132 | 0.0024 |
| | 8 μL | 0.136 | 0.129 | 0.134 | 0.133 | 0.0020 |
| | 10 μL | 0.130 | 0.135 | 0.310 | 0.191 | 0.0591 |
| | BHI + Stem. | 0.231 | 0.314 | 1.291 | 0.512 | 0.2406 |
| *Escherichia coli* | 2 μL | 0.937 | 0.935 | 0.938 | 0.936 | 0.0008 |
| | 4 μL | 0.901 | 0.905 | 0.908 | 0.904 | 0.0020 |
| | 8 μL | 0.610 | 0.615 | 0.607 | 0.610 | 0.0023 |
| | 10 μL | 0.602 | 0.598 | 0.593 | 0.593 | 0.0064 |
| | BHI + Stem. | 1.054 | 1.049 | 1.051 | 1.051 | 0.0014 |

Note: BHI = Brain Heart Infusion broth; $\bar{x}$ = Arithmetic Mean; SEM = (Standard Error of the Mean).

### 3.1. Streptococcus pyogenes

Juniper oil in volumes of 2, 4, 8, and 10 μL, respectively, inhibited the development of *Streptococcus pyogenes*. Statistically, increasing the volume of juniper essential oil to 10 μL led to a reduction in bacterial cell density from 0.786 ± 0.005 to 0.347 ± 0.003, a value much higher than that of the control's, as represented in Figure 1.

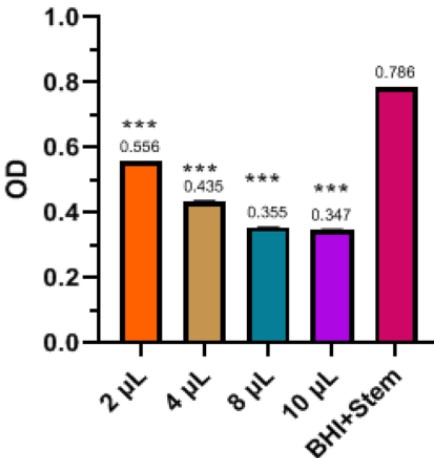

**Figure 1.** Comparative statistical and mean values for juniper oil tested for inhibition of *Streptococcus pyogenes* strain (\*\*\* = *p* < 0.001).

The optical activity of the control was considered to be the potency rate of 100%, and, compared to this in the case of the oil tested by us, we can observe the inhibition of the development of *Str. pyogenes* of 55.81% compared to the control.

### 3.2. Staphylococcus aureus

The optical activity of juniper essential oil on the *Staphylococcus aureus* strain was favorable, with the bacterial cell density after contact with the oil being much lower compared to control (Figure 2). The bacterial density of *S. aureus* culture was also greatly reduced when applying all volumes of essential oil compared to the control (74.54%).

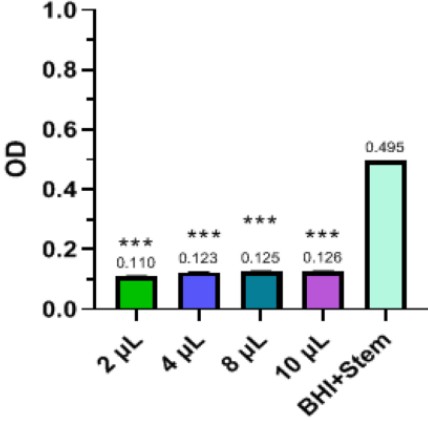

**Figure 2.** Comparative statistical and mean values for juniper oil tested for inhibition of *Staphylococcus aureus* (\*\*\* = *p* < 0.001).

In the *S. aureus* case, the value of OD = 0.110 was found at a dose of 2 μL, but the values of OD reduction for 4, 8, and 10 μL were almost identical (0.123/4 μL, 0.125/8 μL, and 0.126/10 μL, respectively), thus demonstrating the excellent efficiency of WRC *Juniper communis* at all concentrations used in this case.

### 3.3. Pseudomonas aeruginosa

In the case of *Pseudomonas aeruginosa* strain, compared to the control, the bacterial density was reduced to 10 μL essential oil when the bacterial density was reduced by 31.30%; compared to the control, to 8 μL, by 21.70%; and at 4 μL by 21.61%. The bacterial density decreased compared to the control and in direct correlation with the increase in the volume of essential oil applied (Figure 3).

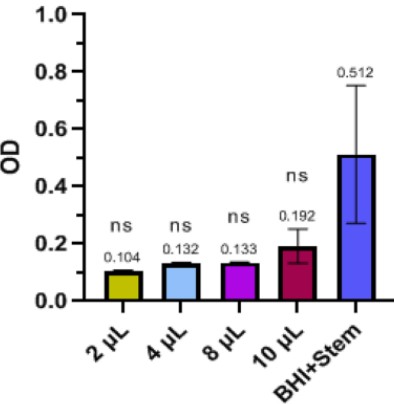

**Figure 3.** Comparative statistical and mean values for juniper oil tested for inhibition of *Pseudomonas aeruginosa* (ns = not significant).

*3.4. Escherichia coli*

Juniper essential oil also reduced bacterial density in the case of *Escherichia coli* strain. The bacterial density of the *E. coli* was reduced, compared to the control, in direct correlation with the increase of the applied volume of juniper essential oil. This was reduced by 43.52% to a volume of 10 μL and by 10.88% to an applied volume of 2 μL, as shown in Figure 4.

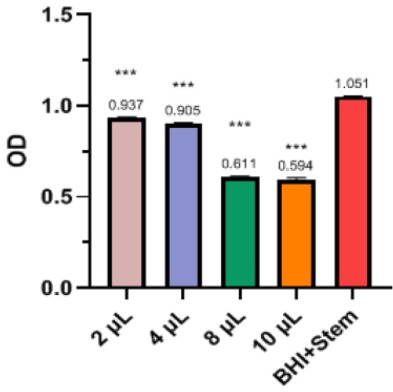

**Figure 4.** Comparative statistical and mean values for juniper oil tested for inhibition of *Escherichia coli* (*** = $p < 0.001$).

**4. Discussion**

In the current study, we attempted to obtain new and valid data about the structure and activity of bioactive compounds provided from Western Romanian Carpathians *Juniperus communis*, which confirmed to be distinctive from others, especially in the case of pinene structures (which, in our case, were superior).

Today there comparative studies associated with the composition and activity of juniper essential oils from different countries are available, and most investigators have shown that there are noteworthy differences in their chemical composition, dissimilarities that are due to soil type, climate, precipitation, temperature, and time of harvest [25].

The most noticeable differentiation was related to the content of pinene structures, which varied high and low, when compared with previous data [5,25,31].

In this respect, we tried to fulfill the lack of information, observing that, in the case of WRC *Juniper communis*, the most elevated concentrations of the major four volatile compounds identified were for β-pinene (34.02%), 1α-pinene (30.43%), p-cymol (20.25%), and β-myrcene (10.20%); the results added new data to those of other investigators of this topic, who have revealed that juniper oil is composed mostly of monoterpene hydrocarbons, most frequently α-pinene, β-pinene, sabinene, mirene, limonene, fenchene, and carene [20].

However, the presence of the other classes' structures, such as hydrocarbons and oxygenated monoterpens, can also have an active role and contribute to the biological activity of juniper berries EOs [31]. This was also observed by Peruč et al. [13,15], who established that the inhibitory effect of juniper EO adjacent to *Mycobacterium* spp. could be due to the abovementioned monoterpene hydrocarbons that can certainly pass the lipidic bi-layer, causing injury to cells.

In a comprehensive study, Falcao et al. [5] identified a total of 97 compounds in juniper OE from Portugal. The EO was characterized and revealed a high content in α-pinene (41.6%), followed by β-pinene (27.6%), but different comparatively with our investigation in Romanian EO, where the obtained percentage was lower for α-pinene, but higher for β-pinene. The synergic activity of the identified components in Romanian juniper, and especially for β-pinene (34.02%) and 1α-pinene (30.43%), was also evaluated as MIC with comparable efficacy in the case of *S. aureus, C. albicans, MRSA,* and *A. baumani* [22].

Moreover, the obtained results allowed us to state that the antibacterial activity of WRC juniper's essential oil is more elevated for Gram-positive pathogens than for the Gram-negative ones. Compared to the control, the reduction in bacterial density was significantly higher for all volumes of essential oils used, but especially for the 10 μL volume, with peak reduction of *Staphylococcus aureus* strain (74.54%), recommending the Romanian essential oil as a beneficial antibacterial resource with multifarious applications.

The results are comparable to those of other researchers in the field, who claim that the essential oil of *Juniper communis* was active against *Escherichia coli* and *Staphylococcus aureus,* but not *Pseudomonas aeruginosa,* which seems to be resistant to this essential oil [11,21,24].

## 5. Conclusions

Following the investigation on the bio-composition and antibacterial efficacy of Western Romanian Carpathians, *Juniper communis* essential oil, we can state that he main four volatile compounds identified were β-pinene, 1α-pinene, p-cymol, and β-myrcene.

Essential oils progressively diminished the bacterial density for all strains tested, but in contrast to Gram-negative bacteria, a much higher antibacterial effectiveness was shown for the Gram-positive pathogens, with the peak reduction in bacterial density being observed for *Staphylococcus aureus* (74.54%).

**Supplementary Materials:** The following supporting information can be downloaded at: https://www.mdpi.com/article/10.3390/app12062949/s1, Figure S1: Gas chromatography–mass spectrometry (GC–MS) of the main profiles of commercial Juniper essential oil from Western Romanian Carpathians.

**Author Contributions:** Conceptualization, E.D. and R.T.C.; methodology, E.A.M. and R.T.C.; software, F.M. and C.S.D.; validation, S.A.O.-A., A.O.D. and R.F.M.; formal analysis, E.A.M.; investigation, E.D., E.A.M. and R.F.M.; resources, C.S.D.; data curation, R.T.C., A.O.D., S.A.O.-A. and F.M.; writing—original draft preparation, E.D. and R.T.C.; writing—review and editing, R.T.C.; visualization, E.D. and F.M.; supervision, R.T.C.; project administration, E.D. and R.T.C.; funding acquisition, R.T.C. All authors have read and agreed to the published version of the manuscript.

**Funding:** This research was funded by the project assigned to the BUASVM from Timisoara through the Institutional Development Fund of the National Council for Financing in Higher Education, 2021 (Grant CNFIS-FDI-2021-0309).

**Institutional Review Board Statement:** Not applicable.

**Informed Consent Statement:** Not applicable.

**Data Availability Statement:** Not applicable.

**Acknowledgments:** This paper is in the frame of COST Action CA18217—European Network for Optimization of Veterinary Antimicrobial Treatment and National Funding application, PN-III-ID-PCE-2021-3.

**Conflicts of Interest:** The authors declare no conflict of interest.

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
