# Peer review of "Juniper communis L. Essential Oils from Western Romanian Carpathians: Bio-Structure and Effective Antibacterial Activity"

_applsci, doi:10.3390/app12062949_

Round 1

Reviewer 1 Report

The experiments are solidly set up and elaborated. The impact of microorganisms may also need to be investigated when multiple species are present, in order to gain a better insight into the antibacterial activity.

Author Response

Point by Point Response to our Reviewers

Reviewer no. 1

Comments and Suggestions for Authors

Reviewer’s comments:

  1. The experiments are solidly set up and elaborated. The impact of microorganisms may also need to be investigated when multiple species are present, in order to gain a better insight into the antibacterial activity.

Answer: Many thanks for the fair & professional approach and also for the encouraging judgment of our work! We have read thoroughly and corrected the manuscript into develop in the Introduction part the impact of microorganisms from our study

Reviewer 2 Report

The manuscript from Eugenia Dumitrescu describes the identification of oils from J. communis and their antibacterial properties.

The article is suitable for applied science and the instrumentation used is state-of-the-art.

However, there are few flaws surrounding this piece of work.

I suggest adding more information on the instrument used. In the text mass spectrometry is reported as detector but there is no information on the parameters. Which detector? Quadrupole? How were the data acquired? Full scan MS/MS?

I also suggest adding few sentences on the identification criteria. The authors described that the compounds were matched with a library. Was retention time used? Was the mass (m/z) of the compound used? Which criteria were considering?

I recommend adding the chromatogram as supplementary figure.

Please provide Figure 2 and figure 3 with better resolution

Please mind the word that hast to be in ‘italic ‘

Author Response

Point by Point Response to our Reviewers

Reviewer no. 2

Comments and Suggestions for Authors

 Reviewer’s comments:

The manuscript from Eugenia Dumitrescu describes the identification of oils from J. communis and their antibacterial properties. The article is suitable for applied science and the instrumentation used is state-of-the-art.

However, there are few flaws surrounding this piece of work.

Answer: Many thanks for the fair & professional approach and also for the encouraging judgment of our work!

 I suggest adding more information on the instrument used. In the text mass spectrometry is reported as detector but there is no information on the parameters. Which detector? Quadrupole? How were the data acquired? Full scan MS/MS?

Answer: Many thanks for the justified observation. We added and corrected all information about methodology used. Please verify!

 I also suggest adding few sentences on the identification criteria. The authors described that the compounds were matched with a library. Was retention time used? Was the mass (m/z) of the compound used? Which criteria were considering?

Answer: Many thanks for the pertinent question. We added all information necessary to be more. Please verify!

 I recommend adding the chromatogram as supplementary figure.

Answer: We added the recommended chromatogram as Supplementary figure 1 in the manuscript

 Please provide Figure 2 and figure 3 with better resolution

Answer: We added more dpi to the figures 2 and 3, now they are of 600 dpi.

Please mind the word that hast to be in ‘italic‘

Answer: We corrected many thanks!

Reviewer 3 Report

  1. Please, mention the abbreviations on the first appearance.
  2. Abstract: Please reconstruct the entire paragraph for a good flow in reading. Line 30: What are the 'obtained results', please elaborate.
  3. Introduction: Please try to mention the novelty in the work. The gap that the work wants to fill. It should not be another repetition of the routine work.
  4. Results: What is SEM? Please separate the tables and figures. It will also be appreciated if a concise figure is used with mean values on the top of the bars.
  5. Discussion: Line 192-195, should be in the results. Line 196-199, doesn't support the figures. It should be clearly reasoned why there is a difference in the action? Why the lowest dose is more effective against Streptococcus and Staphylococcus?
  6. Overall: the study must include other parameters like marker-assisted selection for antimicrobial resistance. The presented report is preliminary to be accepted at this level.

Author Response

Point by Point Response to our Reviewers

Reviewer no. 3

Comments and Suggestions for Authors

 Reviewer’s comments:

Please, mention the abbreviations on the first appearance.

Answer: We verified and corrected in manuscript. Please verify!

Abstract: Please reconstruct the entire paragraph for a good flow in reading. Line 30: What are the 'obtained results', please elaborate.

Answer: We reconstructed and corrected accordingly. Please verify!

Introduction: Please try to mention the novelty in the work. The gap that the work wants to fill. It should not be another repetition of the routine work.

Answer: We introduced info about novelty and necessity of our study. Please verify!

 Results: What is SEM? Please separate the tables and figures. It will also be appreciated if a concise figure is used with mean values on the top of the bars.

Answer: We verified all and corrected accordingly recommendation in text, tables, figures and labels. Please verify!

Discussion: Line 192-195, should be in the results. Line 196-199, doesn't support the figures. It should be clearly reasoned why there is a difference in the action? Why the lowest dose is more effective against Streptococcus and Staphylococcus?

Answer: We verified, corrected and explained in the manuscript. In the case of those two strains the largest, not the lowest dose were effective in Streptococcus case, the lowest optical density was registered at 10 μL = 0.347, comparatively with 2μL = 0.556. In Staphylococcus case, really the value of OD = 0.110 was found to dose of 2 μL, but the values of OD reduction for 4, 8, and 10 μL where almost identical (0.123 / 4 μL, 0.125 / 8 μL and respectively 0.126 / 10 μL, demonstrating the excellent efficiency of WRC (West Romanian Carpathians) Juniper at all concentrations in this case.

Overall: the study must include other parameters like marker-assisted selection for antimicrobial resistance. The presented report is preliminary to be accepted at this level.

Answer: Yes we agree with your pertinent observation, but this study was solely a bio-chemical and an antibacterial activity study, where our aim was to begin with ascertaining the compounds (specifically for Western Carpathians J. communis, where there are no data, please verify the information provided by the main-stream), and which are able to act against the four bacterial strains, mainly encountered in humans and animals in our country. The molecular investigation will follow *(our collective applied for a national research grant), where these aspects will be fulfilled in detail.  Many thanks for the fair & professional approach and also for the encouraging judgment of our work!

Reviewer 4 Report

In publication composition and antimicrobial activity of Juniperus communis EO was described. This publication seems to be within the scope of journal. However it needs several corrections to be more acceptable for publication.

  1. In the introduction, the part about the state of knowledge on the antimicrobial activity of Juniperus communis EO, should be expanded. Compare at least list of references cited in work: Elshafie, H.S., Caputo, L., De Martino, L., Gruľová, D., Zheljazkov, V.Z., De Feo, V. and Camele, I., 2020. Biological investigations of essential oils extracted from three Juniperus species and evaluation of their antimicrobial, antioxidant and cytotoxic activities.Journal of Applied Microbiology129(5), pp.1261-1271.; Peruč, D., Tićac, B., Broznić, D., Maglica, Ž., Šarolić, M. and Gobin, I., 2022. Juniperus communis essential oil limit the biofilm formation of Mycobacterium avium and Mycobacterium intracellulare on polystyrene in a temperature-dependent manner. International Journal of Environmental Health Research32(1), pp.141-154.; Peruč D, Gobin I, Abram M, Broznić D, Svalina T, Štifter S, Staver MM, Tićac B. 2018. Antimycobacterial potential of the juniper berry essential oil in tap water. Arh Hig Rada Toksikol. 69(1):46–54. Peruč D, Tićac B, Abram M, Broznić D, Štifter S, Staver MM, Gobin I. 2019. Synergistic potential of Juniperus communis and Helichrysum italicum essential oils against nontuberculous mycobacteria. J Med Microbiol. 68(5):703–710.
  2. At the end of the introduction, the novelty aspect of presented work should be well presented, because scientific literature about antimicrobial activity of communis EO is reach.
  3. How was the isolation of the communis EO carried out? Was an internal standard (e.g. undecanone) used during isolation? Information of distillation temperature and kind of condenser is needed.
  4. Were standards of at least main compound used in the analysis of the EO composition? Has the GC-MS calibration been performed on the reference hydrocarbon mixture?
  5. Analysis of EO composition (table 1) is absolutely insufficient. Compare with publication: Falcão, S., Bacém, I., Igrejas, G., Rodrigues, P.J., Vilas-Boas, M. and Amaral, J.S. Chemical composition and antimicrobial activity of hydrodistilled oil from juniper berries. Industrial Crops and Products,2018, 124, 878-884. The biological activity of the standard mixtures of major components has so far not so high as original EO, because ingredient, that is present in a trace amount, may be critical to EO activity.
  6. Why the authors have written about atioxidant activity when this activity was not researched in publication? What is the specific relationship between the antioxidant activity of communis EO and the antimicrobial activity? This relationship should be explained thoroughly.
  7. Line 186: Polyphenols are not present in the EO. Information on the isolation of polyphenols from communis can only be found in the introduction.
  8. There is no detailed discussion of the obtained results with previous studies and no emphasis on where the results obtained in this work are better than those described so far. Which components of EO are responsible for this activity and why? What is the mechanism?
  9. In whole manuscript Latin name of microorganisms should be written in italic. Please check carefully whole manuscript. E.g. lines 19 and 21. Please correct evident mistake.

Author Response

Point by Point Response to our Reviewers

Reviewer no. 4

Comments and Suggestions for Authors

In publication composition and antimicrobial activity of Juniperus communis EO was described. This publication seems to be within the scope of journal. However it needs several corrections to be more acceptable for publication.

In the introduction, the part about the state of knowledge on the antimicrobial activity of Juniperus communis EO, should be expanded. Compare at least list of references cited in work:

Elshafie, H.S., Caputo, L., De Martino, L., Grulová, D., Zheljazkov, V.Z., De Feo, V. and Camele, I., 2020. Biological investigations of essential oils extracted from three Juniperus species and evaluation of their antimicrobial, antioxidant and cytotoxic activities. Journal of Applied Microbiology, 129(5), pp.1261-1271.

Peruč, D., Tićac, B., Broznić, D., Maglica, Ž., Šarolić, M. and Gobin, I., 2022. Juniperus communis essential oil limit the biofilm formation of Mycobacterium avium and Mycobacterium intracellulare on polystyrene in a temperature-dependent manner.  International Journal of Environmental Health Research32(1), pp.141-154.

Peruč D, Gobin I, Abram M, Broznić D, Svalina T, Štifter S, Staver MM, Tićac B. 2018. Antimycobacterial potential of the juniper berry essential oil in tap water. Arh Hig Rada Toksikol. 69(1):46–54.

Peruč D, Tićac B, Abram M, Broznić D, Štifter S, Staver MM, Gobin I. 2019. Synergistic potential of Juniperus communis and Helichrysum italicum essential oils against nontuberculous mycobacteria. J Med Microbiol. 68(5):703–710.

Answer: We rephrased and introduced all titles recommended into increase the information value provided by our manuscript. All these papers are novel, and *(really) important, in the context of our manuscript. Many thanks for the good suggestion, valued colleagues!

At the end of the introduction, the novelty aspect of presented work should be well presented, because scientific literature about antimicrobial activity of communis EO is reach.

Answer: We rephrased accordingly this part. Please verify!

How was the isolation of the communis EO carried out? Was an internal standard (e.g. undecanone) used during isolation? Information of distillation temperature and kind of condenser is needed. Were standards of at least main compound used in the analysis of the EO composition? Has the GC-MS calibration been performed on the reference hydrocarbon mixture?

Answer: The oil used was gathered from an authorized facility and it was not extracted by us. Data about CG-MS are also introduced in the manuscript.

 Analysis of EO composition (table 1) is absolutely insufficient.

Compare with publication: Falcão, S., Bacém, I., Igrejas, G., Rodrigues, P.J., Vilas-Boas, M. and Amaral, J.S. Chemical composition and antimicrobial activity of hydrodistilled oil from juniper berries. Industrial Crops and Products, 2018, 124, 878-884. The biological activity of the standard mixtures of major components has so far not so high as original EO, because ingredient, that is present in a trace amount, may be critical to EO activity.

Answer: Yes, a comprehensive table, but here the goal was other. Tough, we added more info referring to principal components analysis (our goal) find by us in Western Carpathians juniper essential oil, also we added this author to the reference list since that table has to be cited. About the main components that we presented, we can guarantee that, at least for S. aureus, the values obtained for all concentration used were certainly efficient from 2 µg to 10 µg. We will develop this study also in a molecular one to prove our findings (as mentioned to another reviewer, we applied for a national grant to tackle this problem).

Why the authors have written about atioxidant activity when this activity was not researched in publication? What is the specific relationship between the antioxidant activity of communis EO and the antimicrobial activity? This relationship should be explained thoroughly.

Answer: We corrected accordingly in the manuscript. Please verify!

Line 186: Polyphenols are not present in the EO. Information on the isolation of polyphenols from communis can only be found in the introduction.

Answer: We corrected accordingly. Please verify!

 There is no detailed discussion of the obtained results with previous studies and no emphasis on where the results obtained in this work are better than those described so far. Which components of EO are responsible for this activity and why? What is the mechanism?

Answer: We rephrased accordingly. Please verify!

 In whole manuscript Latin name of microorganisms should be written in italic. Please check carefully whole manuscript. E.g. lines 19 and 21. Please correct evident mistake.

Answer: We corrected all manuscript including here the Latin names check. Please verify!

Round 2

Reviewer 3 Report

The authors have modified the paper significantly. Although there is enough space for further research in this domain, this paper may contribute as an important database for the future.

Author Response

Rewiever 3 - Comments and Suggestions for Authors

The authors have modified the paper significantly. Although there is enough space for further research in this domain, this paper may contribute as an important database for the future.

Answer: Dear reviewer, many thanks for your work into help us to make our manuscript more attractive for the Journals’ readers! We understood your point and consequently, we added a phrase accordingly in the manuscript.

Reviewer 4 Report

Although the authors have improved the manuscript, in my opinion, it still contains mistakes.

  1. In response to my review, the authors wrote: “The oil used was gathered from an authorized facility and it was not extracted by us.” This statement makes it necessary to rewrite the "Materials and Methods" section. Authors cannot write about distillation in style, which suggests that they obtained the essential oil. It must be made clear that the essential oil came from a specific company with the specific batch number. Although the producers usually guarantee the composition of the essential oil, we observed in detail the variability of the composition depending on the specific batch of the product between the years of our research.
  2. Line 24: It should be “10-3” instead of “10-3”.
  3. Line 84: „naturist” or „natural”? Please remove second „from”.
  4. Latin name of organisms still is not written in italicg. in lines: 33, 78, 163, 247.
  5. Line 133: It should be “(1.5 × 108 CFU × mL-1)” instead of (1.5 × 108 CFU × mL-1)
  6. Line 228: “spp.”should not be written in italic.

Author Response

Point by Point Response to our Reviewers

Rewiever 4 - Comments and Suggestions for Authors

In response to my review, the authors wrote: “The oil used was gathered from an authorized facility and it was not extracted by us.” This statement makes it necessary to rewrite the "Materials and Methods" section. Authors cannot write about distillation in style, which suggests that they obtained the essential oil. It must be made clear that the essential oil came from a specific company with the specific batch number. Although the producers usually guarantee the composition of the essential oil, we observed in detail the variability of the composition depending on the specific batch of the product between the years of our research.

Answer: Dear reviewer, we understand your point and consequently, we added and corrected accordingly to be clear with batch number and year of fabrication. In the M&M part there are presented only: GC-MS identification of volatile compounds, Determination of antimicrobial activity and Statistic analysis and no any info about the oil extraction methodology.

Many thanks for your work into help us to make our manuscript more attractive for the Journals’ readers!

Line 24: It should be “10-3” instead of “10-3”.

Answer: we corrected accordingly

Line 84: „naturist” or „natural”? Please remove second „from”.

Answer: we corrected accordingly all proposition

Latin name of organisms still is not written in italicg. in lines: 33, 78, 163, 247.

Answer: we corrected accordingly

Line 133: It should be “(1.5 × 108 CFU × mL-1)” instead of (1.5 × 108 CFU × mL-1)

Answer: we corrected accordingly. Many thanks observing!

Line 228: “spp.”should not be written in italic.

Answer: we corrected accordingly in manuscript.
